# Proof of concept for multiple nerve transfers to a single target muscle

Matthias Luft[1,2], Johanna Klepetko[1,2], Silvia Muceli[3], Jaime Ibáñez[4,5,6], Vlad Tereshenko[1,2], Christopher Festin[1,2], Gregor Laengle[1,2], Olga Politikou[1,2], Udo Maierhofer[1,2], Dario Farina[4,5], Oskar C Aszmann[1,7], Konstantin Davide Bergmeister[1,2,8]*

[1]Clinical Laboratory for Bionic Extremity Reconstruction, Department of Plastic, Reconstructive and Aesthetic Surgery, Medical University of Vienna, Vienna, Austria; [2]Center for Biomedical Research, Medical University of Vienna, Vienna, Austria; [3]Department of Electrical Engineering, Chalmers University of Technology, Gothenburg, Sweden; [4]Department of Bioengineering, Imperial College London, London, United Kingdom; [5]Department of Clinical and Movement Neuroscience, University College London, London, London, United Kingdom; [6]BSICoS Group, IIS Aragón, Universidad de Zaragoza, Zaragoza, Spain; [7]Department of Plastic, Reconstructive and Aesthetic Surgery, Medical University of Vienna, Vienna, Austria; [8]Karl Landsteiner University of Health Sciences, Department of Plastic, Aesthetic and ReconstructiveSurgery, University Hospital St. Poelten, St. Poelten, Austria

*For correspondence:
kbergmeister@gmail.com

Competing interest: The authors declare that no competing interests exist.

**Abstract** Surgical nerve transfers are used to efficiently treat peripheral nerve injuries, neuromas, phantom limb pain, or improve bionic prosthetic control. Commonly, one donor nerve is transferred to one target muscle. However, the transfer of multiple nerves onto a single target muscle may increase the number of muscle signals for myoelectric prosthetic control and facilitate the treatment of multiple neuromas. Currently, no experimental models are available. This study describes a novel experimental model to investigate the neurophysiological effects of peripheral double nerve transfers to a common target muscle. In 62 male Sprague-Dawley rats, the ulnar nerve of the antebrachium alone (n=30) or together with the anterior interosseus nerve (n=32) was transferred to reinnervate the long head of the biceps brachii. Before neurotization, the motor branch to the biceps' long head was transected at the motor entry point. Twelve weeks after surgery, muscle response to neurotomy, behavioral testing, retrograde labeling, and structural analyses were performed to assess reinnervation. These analyses indicated that all nerves successfully reinnervated the target muscle. No aberrant reinnervation was observed by the originally innervating nerve. Our observations suggest a minimal burden for the animal with no signs of functional deficit in daily activities or auto-mutilation in both procedures. Furthermore, standard neurophysiological analyses for nerve and muscle regeneration were applicable. This newly developed nerve transfer model allows for the reliable and standardized investigation of neural and functional changes following the transfer of multiple donor nerves to one target muscle.

## Introduction

Nerve transfers offer a variety of therapeutic possibilities in modern extremity reconstruction, such as treating peripheral nerve injuries, neuromas, phantom limb pain, improving prosthetic control, or restoring function following spinal cord injuries (*Aszmann et al., 2015*; *Farina et al., 2017*; *Dumanian et al., 2019*; *Van Zyl et al., 2019*). Compared to conventional nerve repair modalities, nerve transfers are capable of bypassing slow peripheral nerve regeneration (*Terzis and Papakonstantinou, 2000*),

thus preventing irreversible muscle fibrosis before reinnervation (**Mackinnon and Novak, 1999**). For this purpose, nearby nerves with a sufficient axonal load and lesser functional importance are neurotomized and transferred to the injured nerve (**Oberlin et al., 1994**; **Bertelli et al., 1997**). Because of overall faster regeneration and better functional outcomes compared to nerve grafting, this surgical procedure has been able to improve the devastating effects of peripheral nerve and brachial plexus lesions, which have otherwise often led to long-term health impairment and subsequent socioeconomic costs (**Mackinnon and Novak, 1999**; **Terzis and Papakonstantinou, 2000**; **Bergmeister et al., 2020**). Additionally, they are used in a procedure termed targeted muscle reinnervation (TMR) to improve myoelectric prosthetic control (**Kuiken et al., 2009**; **Kapelner et al., 2016**), treat neuromas or phantom limb pain (**Mioton et al., 2020**). Here, amputated nerves within an extremity stump are transferred to residual stump muscles, thus significantly improving the recording of neural activity about motor intent and the control of myoelectric prostheses. Generally, one donor nerve is transferred to one target muscle head and this concept has been well studied with high clinical success (**Kuiken et al., 2009**; **Aszmann et al., 2015**; **Farina et al., 2017**). The use of multiple nerve transfers to a single target muscle may further enhance TMR surgery. It could provide additional neuroprosthetic signals and overcome certain limitations in neuromuscular interfacing. Furthermore, transferring multiple nerves to a single target muscle may facilitate neuroma treatment, as not only one but multiple nerves are implanted in a muscle graft or residual limb muscle in an extremity stump (**Herr et al., 2021**).

Although several nerve transfer models have been established (**Kuiken et al., 1995**; **Bergmeister et al., 2016**; **Aman et al., 2019b**), none of them has investigated multiple peripheral nerve transfers in the upper extremity. Only one model where multiple donor nerves are used to restore muscle function in the rat hindlimb has been described (**Kuiken et al., 1995**). However, as most nerve injuries occur in the upper extremity, an upper extremity model for experimental investigation of this concept is needed (**Scholz et al., 2009**).

In this study, we propose a surgical nerve transfer model to allow the transfer of multiple donor nerves to a single muscle head and we validate this model in the rat forelimb. This model allows for reliable analyses with all standard neurophysiological investigations of the motor

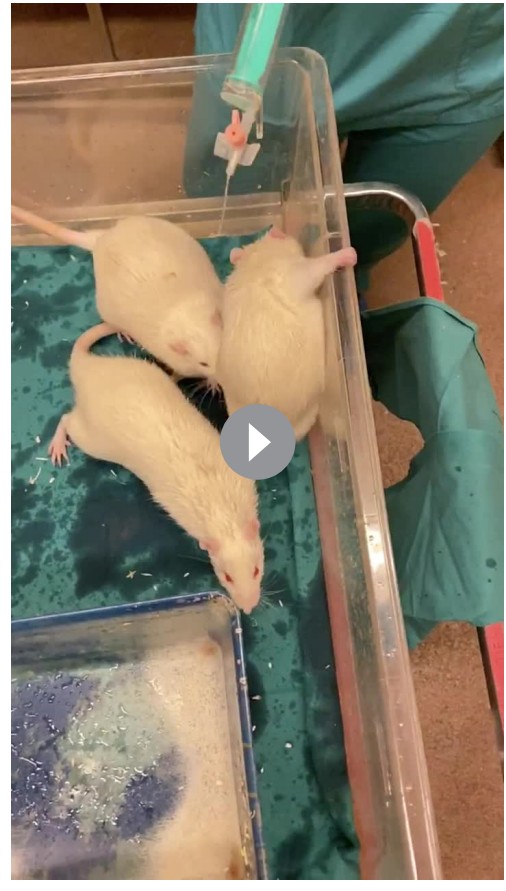

**Video 1.** Grooming behaviour. The grooming behavior of a double nerve transferred animal is provoked by sprinkling 1–3 ml of water or glucose on its snouts and as shown in slow-motion. Notice that the animal can perform a physiological grooming movement with both front paws reaching behind the ears smoothly.
https://elifesciences.org/articles/71312/figures#video1

**Table 1.** Overview of qualitative results.
These results provide a detailed overview of the nerve transfer model and evidence of successful reinnervation.

|  | SNT | DNT |
|---|---|---|
| Surgery time | 49±13 min | 78±20 min |
| Behavior after 12 weeks | All max score (n=21) | All max score (n=30) |
| Macroscopic innervation | All (n=30) | All (n=32) |
| Crush/ neurotomy response | All (n=15) | All (n=17) |

|  | UN | AIN |
|---|---|---|
| Nerve length | 23.08±1.36 mm | 10.50±1.61 mm |

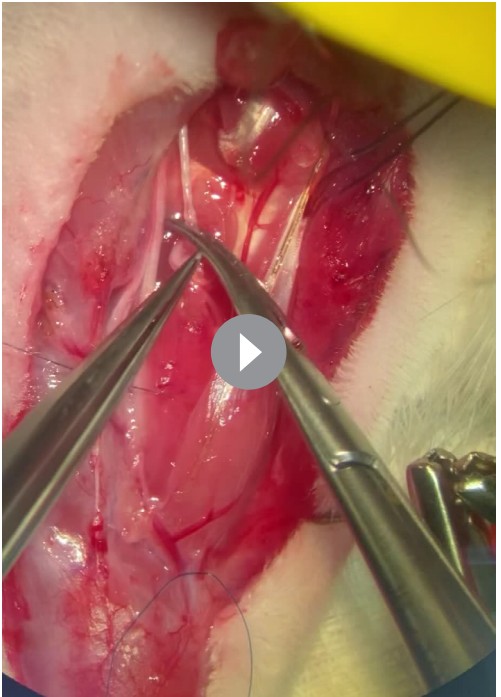

**Video 2.** Nerve crush of the MCN. The supinated left forelimb with the exposed biceps muscle and its motor branch is shown. By crushing the MCN repeatedly with increasing pressure from proximal to distal with a micro needle holder, action potentials were elicited toward the biceps' long head which resulted in muscle fibrillation. MCN, musculocutaneous nerve.
https://elifesciences.org/articles/71312/figures#video2

unit for possible implementation of this concept to clinical application.

## Results
### Nerve transfer surgery
All animals survived the surgical nerve transfers and showed normal gait and grasping behavior in the 12-week follow-up period (*Table 1*). All animals were able to carry out activities of daily behavior unhindered and no signs of severe pain, wound dehiscence, auto-mutilation, or infection were documented. The mean surgery time was 49±13 min for the single nerve transfer (SNT) procedures and 78±20 min for the double nerve transfer (DNT) procedures.

### Behavioral evaluation
Slow-motion video sequence analysis by a blinded evaluator showed that 12 weeks following the SNT and DNT, all animals could consistently reach behind their ears and therefore achieved a maximum score of 5 (*Video 1*, *Video 2*).

### Retrograde labeling
Analyses of the spinal cord following UN transfer showed adequate motor neuron staining in the corresponding segments (Th1-C8). When comparing the spinal cords of the untreated animals with spinal cords of animals that underwent DNT, the distribution pattern of the longitudinally arranged Fluoro-Gold dyed clusters provides strong evidence that both the UN and AIN innervated the biceps' long head (see *Figure 1* for a representative example). Furthermore, no signs of spontaneous regeneration from the musculocutaneous nerve (MCN) were noted by analyzing the corresponding spinal cord segments (C5–C7).

Furthermore, retrograde labeling revealed 50.67±15.67 motor neurons reinnervating the long head of the biceps following SNT and 80.07±28.15 motor neurons following DNT, compared to 67.14±2.34 innervating the untreated biceps (*Figure 1C*).

### Neuromuscular analyses
Both the donor nerve branches and biceps' motor entry point were topographically consistent. The UN measured a mean length of 23.08±1.36 mm from the distal exit of the cubital tunnel to the distal stump. The AIN transfer provided a mean length of 10.50±1.61 mm measured from its branching off the median nerve to the distal stump.

Twelve weeks following nerve transfer surgeries, macroscopic examination of all biceps motor entry points showed successful reinnervation but no auto-innervation by the MCN and no signs of neuroma were detected. Adequate muscle fibrillation was observed in all animals upon crushing and neurotomizing the donor nerves individually following SNT and DNT (AIN crush and UN crush response are shown in *Videos 3 and 4*, respectively).

### Comparison of reinnervated muscle mass
There was a linear relationship between treated and untreated muscle mass for each nerve transfer procedure, as assessed by visual inspection of a scatterplot (*Figure 2—figure supplement 1*). There was homogeneity of regression slopes as the interaction term was not statistically significant, $F$(1,

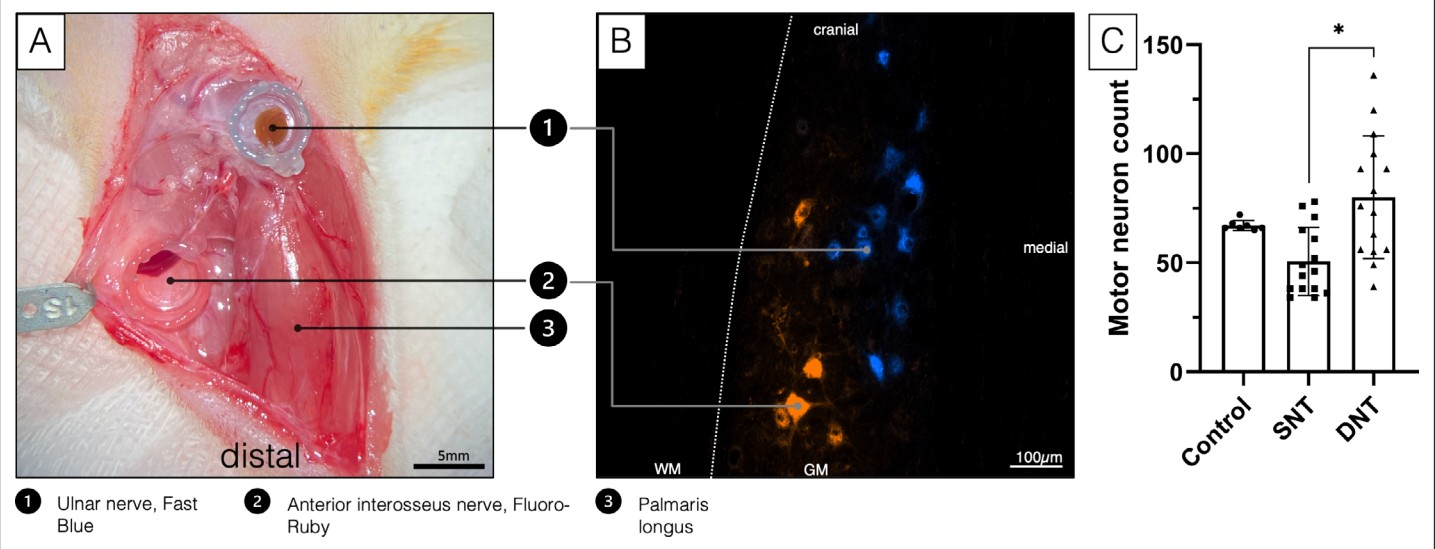

| | |
|---|---|
| **1** Ulnar nerve, Fast Blue | **2** Anterior interosseus nerve, Fluoro-Ruby | **3** Palmaris longus |

**Figure 1.** Double retrograde labeling. (**A**) The selected donor nerves were both dissected in a right forelimb and placed in a conduit reservoir filled with Fast-Blue (UN) and Fluoro-Ruby (AIN), respectively, for 1 hr. Wet sterile swabs were placed above the surgical site to prevent the tissue from drying and the fluorescent dyes from bleaching. (**B**) Spinal cord section C8-Th1. Labeled AIN (orange) and UN motoneuron pool (blue). (**C**) A Kruskal-Wallis H test was conducted to determine if there were differences in labeled motor neuron count between the three groups with different treatment: control (n=7), SNT (n=15), and DNT (n=15). Distributions of motor neuron count were not similar for all groups, as assessed by visual inspection of a boxplot. The mean ranks of motor neuron count were statistically significantly different between groups, $\chi 2(2)=11.147$, p=0.004. Subsequently, pairwise comparisons were performed using Dunn's (1964) procedure with a Bonferroni correction for multiple comparisons. Adjusted p-values are presented. This post hoc analysis revealed statistically significant differences in labeled motor neuron count between the SNT (mean rank=11.90) and DNT (mean rank=24.67; *p=0.004) group, but not between the control group (mean rank=22.07) or any other group combination. DNT, double nerve transfer; GM, grey matter; SNT, single nerve transfer; WM, white matter.

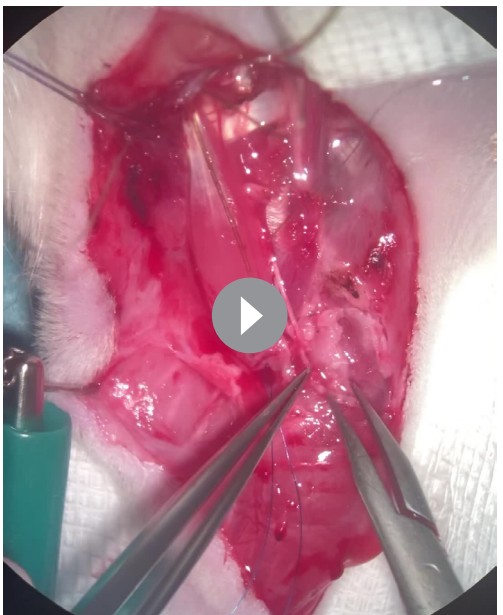

**Video 3.** Nerve crush of the AIN. Twelve weeks following DNT, the AIN reinnervating the long head of the biceps was repeatedly crushed with a micro needle holder. This resulted in a macroscopically recognizable muscle response, indicating successful reinnervation. DNT, double nerve transfer.

https://elifesciences.org/articles/71312/figures#video3

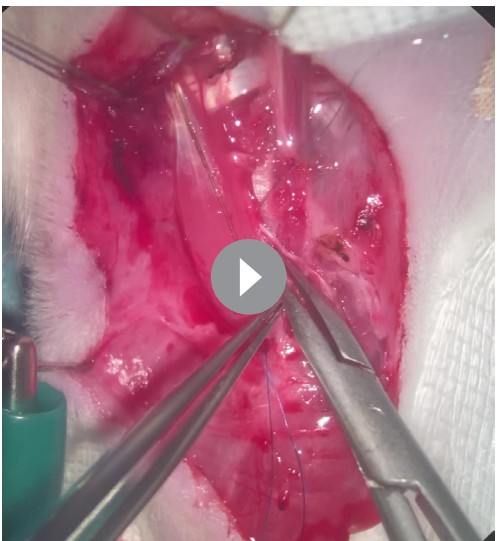

**Video 4.** Nerve crush of the UN. After crushing and neurotomizing the AIN, the UN was crushed. Repeated nerve crushes resulted in adequate muscle fibrillations indicating neuromuscular regeneration.

https://elifesciences.org/articles/71312/figures#video4

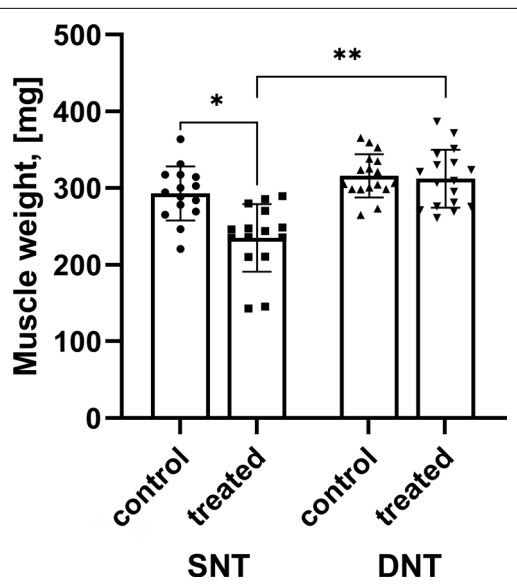

**Figure 2.** Comparison of muscle mass. Muscle mass after SNT was significantly reduced compared to the untreated muscle mass *p<0.001 while muscle mass following DNT regenerated to 98.83%. Muscle mass following DNT was significantly larger compared to the SNT group **p<0.001. DNT, double nerve transfer; SNT, single nerve transfer.

The online version of this article includes the following figure supplement(s) for figure 2:

**Figure supplement 1.** A grouped scatterplot was created to visually assess the linear relationship between treated and untreated muscle mass for each nerve transfer procedure.

28)=0.238, p=0.630. Standardized residuals for the interventions and for the overall model were normally distributed, as assessed by Shapiro-Wilk test (p>0.05). There was homoscedasticity and homogeneity of variances, as assessed by visual inspection of a scatterplot and Levene's test of homogeneity of variance (p=0.504), respectively. There were no outliers in the data, as no cases were detected with standardized residuals greater than ±3 standard deviations.

After adjustment for control muscle mass, there was a statistically significant difference in muscle mass between the treated sides following SNT and DNT, $F_{(1, 29)}=24.030$, ***p<0.001, partial $\eta^2=0.453$ (*Figure 2*) . Muscle mass was statistically significantly larger in the DNT group (303.01±7.76 mg) compared to the SNT group (245.57±8.29 mg), with a mean difference of 57.45 (95% confidence interval [CI], 33.48–81.41) mg, ***p<0.001. Data are reported adjusted mean± standard error.

## Comparison of reinnervated and control muscle mass

No outliers were detected as assessed by inspection of a boxplot. The assumption of normality was not violated, as assessed by Shapiro-Wilk test for the SNT (p=0.758) and DNT groups (p=0.307).

The mean muscle mass was reduced following SNT (235.07±44.05 mg) as opposed to the untreated contralateral side (292.93±35.17 mg) with a statistically significant decrease of –57.87 (95% CI, –77.38 to –38.35) mg, $t_{(14)}=-6.360$, ***p<0.001, d=1.64 (*Figure 2*). However, mean muscle mass following DNT (312.28±37.74 mg) compared to the untreated contralateral side (315.97±28.22 mg) was similar and showed no statistically significant change (p=0.571). Data are reported as mean ± standard deviation.

## Discussion

The present study provides a robust and easily accessible model for surgical DNTs to a single target muscle in the rat's upper extremity. We offer detailed step-by-step instructions on how to reproduce this model, including potential pitfalls. For comparison, the model also offers a description of a SNT to the same target muscle. We employed nerve crush, neurotomy, behavioral analysis, and retrograde labeling which indicated that neuromuscular regeneration of two donor nerves occurred into one target muscle.

To our knowledge, only one rat model for multiple peripheral innervation of a single target has been described. However, that previous model was for the lower extremity and did not provide a detailed description for step-by-step reproduction of the model (*Kuiken et al., 1995*). Hindlimb models do not adequately represent the physiology of upper extremity nerve transfers and TMR procedures. The key differences between upper and lower extremities are amount of usage of the limbs, complexity of movement, weight-bearing, and of course sensorimotor circuitry. This notion is supported by the clinical discrepancy between the excellent outcomes for upper extremity compared to the poor outcomes for lower extremity nerve transfers (*Ray et al., 2016*). Furthermore, most nerve transfers are currently conducted in the upper extremity for both nerve reconstruction and prosthetic

control. We already established single peripheral nerve transfer models in the upper extremity (*Bergmeister et al., 2016*; *Aman et al., 2019b*), which were considered for developing this novel model. For this purpose, we conducted anatomical dissections in eight rat cadavers to design the DNT concept to allow tension-free approximation of the two motor nerves to the target biceps muscle. Theoretically, many other target muscles are also feasible due to the sufficient length of both the UN and AIN. However, the biceps muscle provides an optimal target that is accessible for all standard structural and functional analyses and accurately represents a surgical target in clinical nerve transfer scenarios as well.

The implementation of this model requires an operating microscope, a set of microsurgery tools, and advanced microsurgical skills to achieve reproducible results. In our experience, dissection of the UN in the antebrachium can be performed in a straightforward manner and preservation of the motor branch to the flexor carpi ulnaris muscle, the dorsal sensory branch, and the ulnar artery is easily feasible. Subsequently, transecting the UN as distally as possible allows for tension-free coaptation to the proximal target muscle. Exposure of the MCN's motor branch to the long head of the biceps is best achieved in the bicipital groove by retracting the overlaying pectoral muscles medially. Here, considerable care must be taken when dividing the two bicep heads to preserve the bicipital artery, which enters the long head in the distal portion and advances in proximal direction. Injury to this vessel has shown to affect functional measures in previous experiments. Another hazard in the DNT model is potential injury of the median vessels in the cubital fossa. To prevent this scenario, special attention is required during the dissection of the median nerve, because the median vessels are either found directly beneath or above the nerve. It is mandatory to dissect the AIN intraneurally to its proximal branching point to enable tension-free coaptation to the original motor point of the biceps. Due to the target to donor nerve diameter discrepancies, we chose to suture the donor nerves to the motor entry point epimysially. In previous models, this approach led to reliable reinnervation of the target muscle (*Bergmeister et al., 2019*).

Our behavioral observations indicate that the procedures did not cause extraordinary distress or pain under adequate analgesia postoperatively. As early as 1 week after surgery, behavioral testing was carried out in randomly selected individual animals, and all of them achieved the maximum score. Likewise, after a 12 -week regeneration period, all animals from both the control and the experimental DNT group achieved the maximum score of Terzis grooming test (*Inciong et al., 2000*; *Video 1*). Hence, it seems that two motor nerves of different origins governing the same muscle did not hamper activities of daily living. Additionally, no substantial pain or neuroma pain was evident. When comparing the two procedures, it takes only marginally longer to perform the DNT, while no additional physical stress or motor deficits were observed postoperatively.

The donor nerves reinnervated the target muscle within 12 weeks in all animals as indicated macroscopically during dissection and by the fact that nerve crush or neurotomy induced fasciculations of the muscle (*Videos 3 and 4*). Likewise, intramuscular retrograde labeling showed the uptake and transport of tracer dye into the motor neuron columns of the two transferred nerves. Retrograde labeling further indicated that the overall number of motor units was reduced to 75.47% in SNT but increased to 119.26% in DNT compared to the untreated control side. Most interestingly, there was a statistically significant increase of 58.02% in motor units following DNT compared to SNT. This suggests that DNT may be more likely to sufficiently innervate muscle fibers and furthermore potentially increase myosignals for prosthetic interfacing, as has been previously shown (*Bergmeister et al., 2019*).

Interestingly, after 12 weeks, muscle mass of the UN reinnervated muscles only recovered to 80.25% of the contralateral side. This is in contrast with previous studies performed by the authors of this work (*Bergmeister et al., 2019*). A possible explanation for this mismatch is the difference of the levels at which the UN was cut and transferred in the two studies. Unlike in the previous study where the entire UN was transferred, here the UN was transferred at the wrist level. This may have caused that the donor nerve was not able to fully regenerate the long head of the biceps due to the lower motor axon numbers. Detailed analyses exist for humans, where the UN at wrist level only contains 1226±243 motor axons compared to the entire UN (2670±347) whereas the MCN contains 1601±164 (*Gesslbauer et al., 2017*). Considering that the muscle mass of double reinnervated muscles regenerated to 98.83%, it appears that the two donor nerves were better able to reinnervate and adequately restore 24.72% more muscle mass than the SNT. This additionally indicates that both SNT and DNT

procedures were successful and that DNT with a high axonal load may lead to higher muscle reinnervation and functional regeneration.

Previous findings (*Bergmeister et al., 2019*) reported neuroma formation at the insertion point following nerve transfer. These consisted presumably mainly of sensory axons and the surplus of motor neurons which was not able to innervate motor endplates. We did not observe neuroma formation in this study and believe, that this is because the donor nerves comprised only a few sensory axons and the donor-to-recipient ratio of motor axons and targets was more balanced than in the previous study, as mentioned above. Therefore, we assume that no fibers were lost at the insertion site to the muscle, which may have formed a neuroma. Although the question of the optimal donor-to-recipient ratio for optimal outcome remains unsolved, further investigations in this surgical model are ongoing to answer this question and contribute to the surgical refinement of nerve transfers.

The presented nerve transfer model finds broad application in many research fields. It offers the possibility to investigate basic neurophysiology, but also clinical applications of surgical nerve transfers for biological reconstruction. Additionally, the application of the DNT to already established neuromuscular interfacing approaches for bionic reconstruction could be explored. Regenerative peripheral nerve interfaces RPNIs (*Vu et al., 2020*), for example, are created by implanting transected peripheral nerves into small free muscle grafts, which serve as neuronal control signal amplifiers. Particularly with RPNIs, the DNT could be advantageous as muscle mass is currently constrained by the axonal count and therefore limiting EMG output for neuroprosthetic control. Potentially beneficial applications could also be explored with the agonist-antagonist myoneural interface (AMI) (*Srinivasan et al., 2017*). An AMI consists of a pair of muscles connected to each other by its tendons whereas the contraction of one causing the stretching of the other and vice versa. The contraction and its EMG signals serve as control signal for the prosthesis and the stretch as afferent proprioceptive feedback.

Another method that could most likely benefit from the DNT is TMR (*Kuiken et al., 2009*). After amputation, TMR can create additional myosignals to improve basic prosthetic control. In TMR, neuromas within the stump are cut and the healthy fascicles are then transferred to intact muscle segments, after denervation from their original innervation. Earlier studies revealed that EMG technology can record and decipher neuronal signals from those reinnervated areas into signals for prosthetic movement (*Bergmeister et al., 2017*; *Muceli et al., 2019b*; *Salminger et al., 2019*). The biceps' long head is suitable to perform various EMG examinations, as we have previously shown (*Bergmeister et al., 2019*; *Muceli et al., 2019a*). Especially with novel multichannel EMG technology (*Muceli et al., 2015*), individual motor unit action potentials can potentially be decoded from such signals as we have previously shown in SNT models (*Muceli et al., 2019a*).

We want to emphasize that this was a first attempt at controlled multiple innervation of a single target in a murine model. Furthermore, we would like to point out that the clinical translatability of validated rat models may require the verification of scalability in larger animal models first (*Aman et al., 2019a*). An obvious limitation is the lack of investigation of voluntary motor activity since this is not feasible in rats. It remains unclear whether a human could access a single muscle via two different nerves. Another potential limitation of this study is the use of the mixed UN containing both sensory and motor nerve fibers. For better outcomes of surgical nerve transfers, 'pure' motor nerves should be preferred, such as the AIN used here, to avoid sensory to motor axon incongruence (*Ray et al., 2016*). We decided to transfer the UN at a level, where it also contains sensory fibers of the superficial branch because unlike in humans, intraneural fascicular dissection to identify the two branches proximal to Guyon's canal is impossible due to the intermingling of axons at the level of Guyon's canal. Uncomplicated dissection, significant transfer leeway, and the lack of better alternatives made the UN the best option.

In conclusion, this study demonstrated that a single target muscle can host two separate donor nerves. Our results suggest that both the SNT and DNT models are suitable for common neurophysiological examinations in peripheral nerve research. The concept of transferring multiple nerves to a single target may improve muscle reinnervation, prosthetic interfacing, neuroma therapy or facilitate phantom limb pain management. Until first clinical interfacing applications can be translated, further EMG analysis and validation are needed to fully understand the neurophysiological changes following multiple nerve transfers.

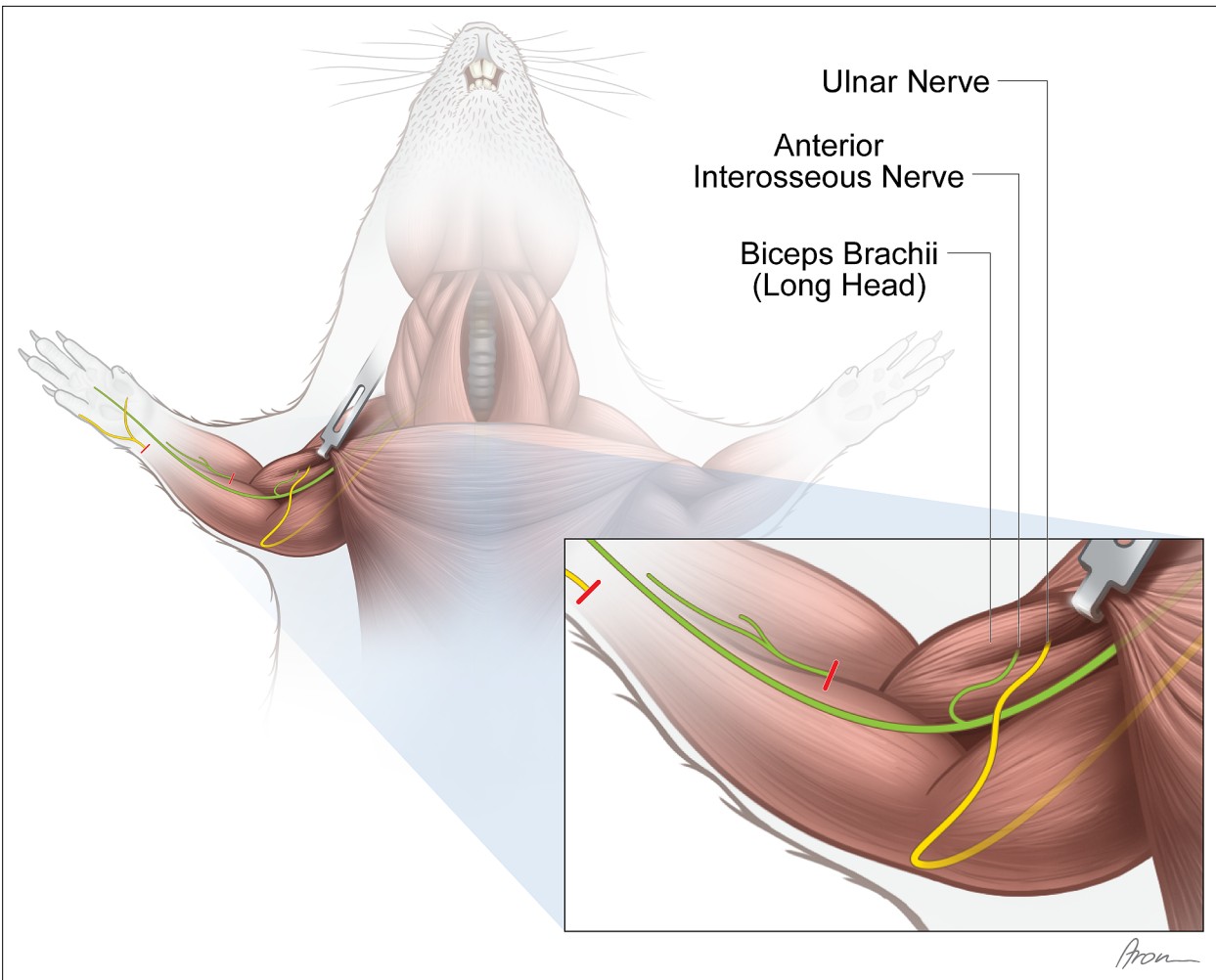

**Figure 3.** Experimental nerve transfer models. *Single-nerve transfer model:* The UN (yellow) was transected distally to the palmar cutaneous branch in the forearm and surgically transferred to reinnervate the long head of the biceps (n=30). *Multiple-nerve transfer model:* Both the UN (yellow) and AIN (green) were redirected to reinnervate the long head of the biceps (n=32). Before both nerve transfer procedures, the originally innervating branch of the MCN was removed. The untreated contralateral biceps muscles served as internal control for both groups. The red lines indicate the level of transection.

Credit: Aron Cserveny. MCN, musculocutaneous nerve.

## Materials and methods
### Experimental design
Eight rat cadavers were dissected to design the DNT procedure. An important criterion for the selection of the donor nerves and the target muscle was clinical relevance. First, eligible peripheral motor nerves were determined for a reliable, tension-free transfer to the long head of the biceps muscle. Then, the topographical relationships between the biceps' long head, its motor nerve branch, the ulnar nerve in the antebrachium (UN) and the anterior interosseus nerve (AIN) were studied and subsequently compared to the human anatomy. These studies verified the anatomical feasibility of transferring both the distal UN and AIN to the long head of the biceps.

Sixty-two Sprague-Dawley rats aged 8–10 weeks were randomly allocated into two groups to investigate functional and structural changes following SNT and DNT. Thirty-two animals were assigned to the DNT group (*Figure 3*), while 30 animals underwent the SNT of the UN and were used as control (*Figure 3*). Twelve weeks after surgery, microscopic inspection of the motor entry point (n=62), nerve crush and neurotomy (n=32), and Terzis' grooming test (n=51) (*Inciong et al., 2000*) were performed. After the final functional assessments, muscle specimens were harvested and weighed (n=32). Forty-five animals were assigned for retrograde labeling analyses. Sample size calculations performed by a

biostatistician were considered in the planning of the studies. Planning, conducting, and reporting of experiments were performed according to the ARRIVE (Animal Research: Reporting of In Vivo Experiments) guidelines (*Percie Du Sert et al., 2020*). The protocols for these experiments were approved by the ethics committee of the Medical University of Vienna and the Austrian Ministry for Research and Science (reference number BMBWF – 66.009/0413 V/3b/2019) and strictly followed the principles of laboratory animal care as recommended by the Federation of European Laboratory Animal Science Associations (FELASA) (*Guillen, 2012*).

## Nerve transfer model

For each procedure, anesthesia was induced with ketamine (100 mg/kg) and xylazine (5 mg/kg) intraperitoneally and maintained by volume-controlled ventilation (40% O2, room air, 1.5–2% isoflurane) following orotracheal intubation. Piritramide (0.3 mg/kg) was administered subcutaneously for analgesia. Furthermore, the drinking water was mixed with piritramide and glucose (30 mg piritramide and 30 ml 10% glucose dissolved in 250 ml drinking water) and administered ad libitum for pain relief during the first 7 postoperative days. After the experimental tests, animals were euthanized with a lethal dose of pentobarbital (300 mg/kg) injected intracardially under deep anesthesia. All animals were examined daily by an animal keeper for pain, sensory deficits, impairments in daily activities, wound dehiscence, and infection. All nerve transfer procedures were performed by the same surgeon and assistant. Nerve transfer models such as the one described here aim to be as reproducible as possible and to be able to modify them for research purposes if necessary.

## Single nerve transfer

A lazy S-shaped incision was made from 5 mm caudal to the greater tubercle of the humerus over the medial epicondyle along the ulnar side of the forearm until 5 mm proximal to the forepaw (*Figure 4A*). Following the dissection of the subcutaneous tissue, the antebrachial fascia was opened through an incision placed over the palmaris longus muscle to preserve the underlying ulnar collateral vessels. Then, the flexor carpi ulnaris muscle was bluntly mobilized and retracted ulnarly using a Magnetic Fixator Retraction System (Fine Science Tools, Heidelberg, Germany) to expose the UN. Further exposure of the dorsal and palmar cutaneous branches of the UN was carried out using an operating microscope (Carl Zeiss, Munich, Germany) (*Figure 4B*). The palmar branch was cut right after its emergence and the UN was subsequently transected as distally as possible. The UN was dissected proximally to its distal exit from the cubital tunnel while preserving the ulnar artery and basilic vein. Intraneural dissection allowed for conservation of the dorsal cutaneous and flexor carpi ulnaris motor branches (*Figure 4B*), while facilitating a tension-free nerve coaptation. Next, the incision of the antebrachial fascia was extended proximally to open the brachial fascia above the cubital fossa and biceps. Subsequently, the pectoral muscles were retracted to expose the MCN branch to the long head of the biceps running along the bicipital groove (*Figure 4C*). The motor branch of the MCN to the biceps' long head was then cut at the motor insertion point and the proximal segment was subsequently removed from its division to prevent spontaneous regeneration. Next, the UN was routed proximally over the cubital fossa and coapted tension-free to the epimysium near the original motor insertion point with one 11–0 (Ethilon, Ethicon, Johnson & Johnson Medical Care, USA) simple interrupted stitch (*Figure 4D*).

## Double nerve transfer

The skin incision, exposure of the distal UN as well as the denervation of the biceps' long head were performed as described in the SNT. Before coaptation of the UN, the median nerve and AIN were dissected. For better exposure of the AIN, one blunt retractor was carefully placed to pull the proximal belly of the pronator teres muscle ulnarly (*Figure 5A*). After identifying the AIN, it was transected and dissected proximally in an intraneural fashion to its branching point (*Figure 5A*). Then, both the UN and the AIN were neurotized to the epimysium near the original motor insertion point with one 11–0 (Ethilon, Ethicon, Johnson & Johnson Medical Care) simple interrupted stitch each (*Figure 5B*). Significant caliber differences between the motor branch of the biceps' long head and the two transferred nerves required neurotization directly to the epimysium. In this way, the regeneration distance was kept as short as possible, hence minimizing the reinnervation time. It is particularly important not to place the two nerves in direct proximity in the tissue (*Figure 5B*) as this increases the complexity of the

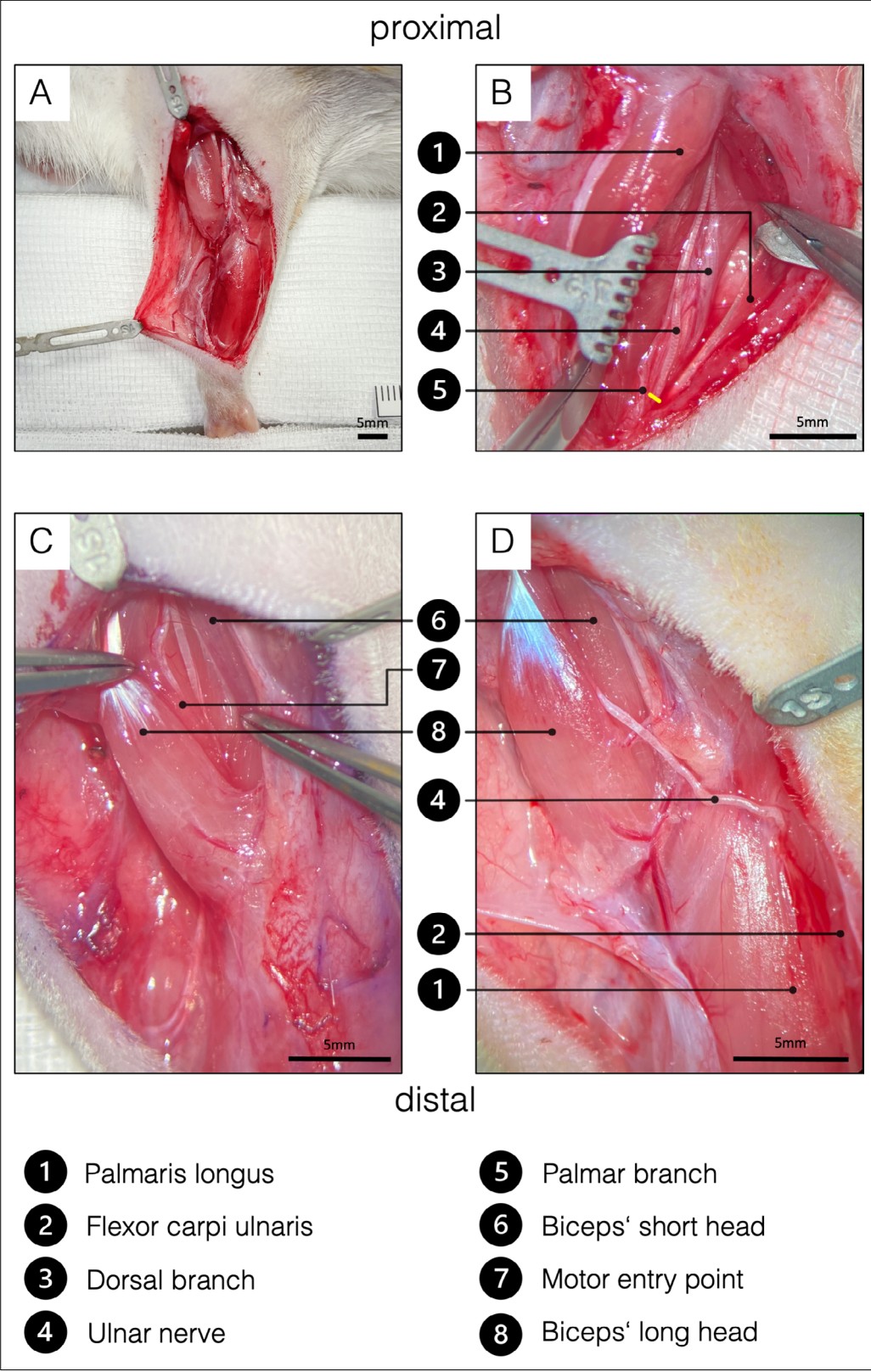

**Figure 4.** Surgical procedure of the ulnar nerve transfer. (**A**) Overview of the rats' supinated right forelimb after the brachial and antebrachial fascia were removed. (**B**) Two blunt retractors have been placed to pull the flexor carpi ulnaris and the palmaris longus apart, revealing the underlying UN. The yellow line indicates the level of transection to gain sufficient length to reach the biceps' long head tension-free. To achieve this, the palmar

*Figure 4 continued on next page*

*Figure 4 continued*

cutaneous branch must be transected, while the dorsal cutaneous branch can be preserved. (**C**) For better visualization, the brachial fascia was opened above the biceps. A sharp retractor was placed to pull back the pectoral muscles and thus revealed the two biceps heads, which were bluntly separated. In the deep bicipital groove, the MCN and its motor branch to the long head of the biceps were identified. Maximum length of the motor branch to the long head was removed to prevent spontaneous regeneration. (**D**) Eventually, the UN was rerouted from between the palmaris longus and flexor carpi ulnaris to the long head of the biceps and sutured to the epimysium at the former original motor entry point. This procedure on the one hand spares the denervation of the flexor carpi ulnaris and the flexor digitorum superficialis and the invasive dissection through the cubital tunnel. MCN, musculocutaneous nerve.

The online version of this article includes the following source data for figure 4:

**Source data 1.** Labeled motor neuron count by treatment.

dissection and therefore the risk of injuring the nerves in the follow-up examinations. Wound closure was performed with fascial and deep dermal 6–0 (Vicryl, Ethicon, Johnson and Johnson Medical Care, Austria) simple interrupted sutures followed by running subcuticular suture with 6–0 (Vicryl, Ethicon, Johnson and Johnson Medical Care).

## Behavioral valuation

Quantitative assessment of grooming behavior was carried out and filmed 12 weeks after the SNT (n=21) and DNT (n=30) using Terzis' grooming test (*Inciong et al., 2000*), a modification of Bertelli's grooming test (*Bertelli and Mira, 1993*). To keep the animals' stress level at a minimum, testing was performed in the animals' familiar environment. In brief, 1–3 ml of water was sprinkled on the

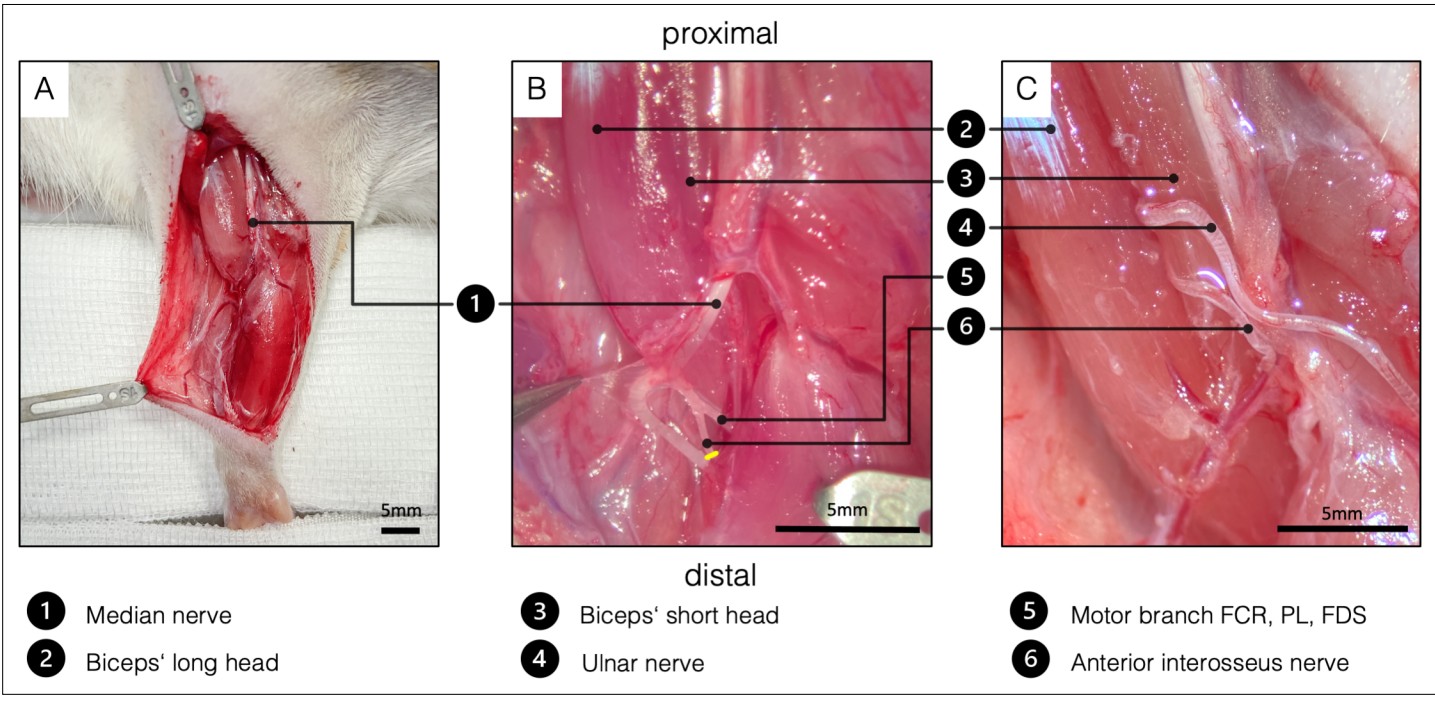

**Figure 5.** Surgical procedure of the double nerve transfer. (**A**) General view of the right supinated forelimb. The proximal hook pulls the pectoral muscles toward proximal for better presentation. (**B**) The brachial and antebrachial fascia and the motor branch to the pronator teres muscle were removed for better visualization. In the cubital fossa, three branches arise from the median nerve: one muscle branch supplying the pronator teres (resected); one muscle branch supplying the flexor carpi radialis, palmaris longus, and flexor digitorum superficialis; and the AIN supplying pronator quadratus, flexor pollicis longus, and flexor digitorum profundus. After transecting the AIN (yellow line), proximal dissection in an intraneural fashion gains sufficient length to reach the biceps' motor entry point. (**C**) Surgical site before wound closure, after both the UN and the AIN were transferred to the physiological motor entry point of the long head of the biceps. FCR, flexor carpi radialis; FDS, flexor digitorum superficialis; PL - palmaris longus.

The online version of this article includes the following source data for figure 5:

**Source data 1.** Muscle mass of treated and untreated sides in [mg] by treatment.

rats' snouts, which led to consistent bilateral grooming movements of the forelimbs. Grading of the grooming performance was assessed by the following score: grade 1, paws reach mouth or elbow is extended; grade 2, paws reach mouth and beneath eyes; grade 3, paws reach eyes; grade 4, paws reach between eyes and ears; and grade 5, paws reach behind the ears. The slow-motion video sequences were graded by a blinded observer.

## Retrograde labeling

Assessment of the motor unit at the spinal cord level after nerve transfer surgery was performed via retrograde labeling as previously described (*Hayashi et al., 2007*). In brief, retrograde tracers are taken up by terminal axons and transported via retrograde axonal transport to label the cell somas in the spinal cords' ventral root. In eight additional untreated control animals, both the UN in the ante-brachium and the AIN were transected and placed into conduit reservoirs for 1 hr, either filled with 5 µl of 10% Fluoro-Ruby (Invitrogen, Carlsbad, CA) or 5 µl of 2% Fast-Blue (Polysciences, Warrington, PA). Tracer leakage was prevented by sealing the reservoir around the nerve with Vaseline (Vaselinum album, Fagron, Glinde, Germany). Hence, the corresponding motor neuron pools in the spinal cord (C8-Th1) were localized (*Figure 1*). To further prevent bias due to differences in penetration of the tracers, the nerves were alternately colored with Fluoro-Ruby and Fast-Blue. Additionally, 12 weeks following SNT (n=15), DNT (n=15), and in another seven untreated control animals, motor neurons reinnervating the long head of the biceps were studied. Through a 15 -mm incision above the biceps, the biceps' long head and its insertion site were exposed. A Hamilton micro syringe was then used to inject 10 µl 2% Fluoro-Gold (Fluorochrome, LLC, Denver, CO) evenly into the biceps' long head near the motor insertion site. After tracer injection with a small gauge needle, the syringe was kept inside the muscle for 1 min before slowly withdrawing it to keep leakage to a minimum. Seven days following retrograde labeling, the animals were deeply anesthetized by a lethal dose of xylazine, ketamine, and pentobarbital intraperitoneally before the left ventricle was perfused with 400 ml of 0.9% NaCl followed by 400 ml of 4% paraformaldehyde (PFA) solution. Then, the spinal cord segments C4-Th2 were harvested and stored in 4% PFA for 24 hr at +4°, followed by 24 hr in 0.1 M phosphate-buffered saline PBS0 at +4°. Then, the specimens were dehydrated in a PBS solution with increasing sucrose concentrations of 10%, 25%, and 40% for 24 hr each before embedding them in Tissue-Tek O.C.T. Compound (Sakura Finetek Europe B.V., Alphen aan den Rijn, Netherlands). Spinal cord segments were cut longitudinally into 40 µm sections using a cryostat (Leica, Germany). To assess the reinner-vation and motor neuron count, each spinal cord section was analyzed in an observer blinded setting using a fluorescence microscope (Carl Zeiss, Munich, Germany). Spinal cord segments after SNT and DNT (Fluoro-Gold) were compared to the double labeled (Fast-Blue, Fluoro-Ruby) and the intramus-cular labeled Fluoro-Gold segments of the untreated animals.

## Neuromuscular analyses

The lengths of both the UN (n=6) and AIN (n=6) were measured intraoperatively before coaptation to the muscle. Twelve weeks following surgery, the motor entry point was microscopically exam-ined for proper reinnervation and neuroma formation in all animals. Muscle reaction to nerve crush (see *Video 2* for muscle reaction to MCN crush in the control side) and neurotomy was assessed in animals following DNT (n=17) and compared to animals following SNT (n=15). For internal control, the motor branches to the biceps' long head were crushed and neurotomized in the contralateral forelimbs. Conclusively, to assess neuromuscular regeneration after denervation, the biceps muscles were resected and weighed immediately after removal using a microscale.

## Statistical analysis

To compare the motor neuron count of retrograde labeling between the three groups, a one-way anal-ysis of variance (ANOVA) would have to be conducted. Since two assumptions (one significant outlier as assessed by inspection of a boxplot and non-normally distributed data as assessed by Shapiro-Wilk test) for the ANOVA have not been met, a Kruskal-Wallis H test was performed instead. Distribution of scores was assessed by visual inspection of a boxplot.

Diagnostic plots were considered to check the assumptions of parametric models. An analysis of covariance (ANCOVA) was conducted to determine the effects of the nerve transfer procedure (SNT and DNT) on the reinnervated muscle mass after adjusting for control muscle mass. It must be

assumed that the control muscle mass may have had an undesirable influence on the treated muscle mass and the ANCOVA takes this influence into account without, however, changing anything in the experiment. Shapiro-Wilk test and Levene test were performed to check for normal distribution of standardized residuals and homogeneity of variances, respectively.

In addition, a paired-samples t-test was used to determine whether there was a change of muscle mass following SNT or DNT between the two sides. All data analyses were performed using SPSS Statistics for Macintosh, version 25.0 (IBM, Armonk, NY).

## Acknowledgements

The authors thank AM Willensdorfer for her continuous technical assistance and A Cserveny for his admirable illustrations in this project. In addition, the authors thank Florian Frommlet for his statistical analyses and expertise as a biostatistician.

## Additional information

### Funding

| Funder | Grant reference number | Author |
|---|---|---|
| European Research Council | ERC Synergy Grant: No 810346 | Matthias Luft<br>Vlad Tereshenko<br>Christopher Festin<br>Gregor Laengle<br>Olga Politikou<br>Udo Maierhofer<br>Dario Farina<br>Oskar C Aszmann<br>Konstantin Davide Bergmeister<br>Jaime Ibáñez<br>Silvia Muceli |
| Chalmers Life Science Engineering Area of Advance | | Silvia Muceli |

The funders had no role in study design, data collection and interpretation, or the decision to submit the work for publication.

### Author contributions

Matthias Luft, Conceptualization, Data curation, Formal analysis, Investigation, Methodology, Project administration, Validation, Visualization, Writing – original draft, Writing – review and editing; Johanna Klepetko, Conceptualization, Data curation, Formal analysis, Methodology, Project administration, Writing – review and editing; Silvia Muceli, Jaime Ibáñez, Conceptualization, Data curation, Formal analysis, Investigation, Methodology, Validation, Writing – original draft, Writing – review and editing; Vlad Tereshenko, Conceptualization, Data curation, Formal analysis, Methodology, Writing – review and editing; Christopher Festin, Conceptualization, Data curation, Formal analysis, Methodology, Validation, Visualization, Writing – review and editing; Gregor Laengle, Conceptualization, Data curation, Formal analysis, Investigation, Validation, Writing – review and editing; Olga Politikou, Conceptualization, Data curation, Formal analysis, Methodology, Validation, Writing – review and editing; Udo Maierhofer, Conceptualization, Formal analysis, Investigation, Methodology, Project administration, Writing – review and editing; Dario Farina, Conceptualization, Formal analysis, Funding acquisition, Methodology, Project administration, Resources, Writing – review and editing; Oskar C Aszmann, Conceptualization, Data curation, Formal analysis, Funding acquisition, Investigation, Methodology, Resources, Supervision, Validation, Writing – review and editing; Konstantin Davide Bergmeister, Conceptualization, Data curation, Formal analysis, Investigation, Methodology, Supervision, Validation, Writing – original draft, Writing – review and editing

### Author ORCIDs

Matthias Luft http://orcid.org/0000-0002-9161-4125

Silvia Muceli http://orcid.org/0000-0002-0310-1021
Vlad Tereshenko http://orcid.org/0000-0001-7761-5191
Gregor Laengle http://orcid.org/0000-0003-1011-3482
Dario Farina http://orcid.org/0000-0002-7883-2697
Konstantin Davide Bergmeister http://orcid.org/0000-0003-3910-9727

## Ethics

The protocols for the experiments were approved by the ethics committee of the Medical University of Vienna and the Austrian Ministry for Research and Science (reference number BMBWF- 66.009/0413-V/3b/2019) and strictly followed the principles of laboratory animal care as recommended by the Federation of European Laboratory Animal Science Associations (FELASA).

## Decision letter and Author response

Decision letter https://doi.org/10.7554/eLife.71312.sa1
Author response https://doi.org/10.7554/eLife.71312.sa2

---

## Additional files

### Supplementary files

• Transparent reporting form

• Source code 1. The Syntax code for SPSS 25 for Mac is divided into analysis for muscle mass on page 1 (ANCOVA) and retrograde labeling analysis (Kruskal Wallis H-Test) on page 2. The respective headings for the code sections are highlighted using asterisks.

### Data availability

Muscle mass data have been deposited in Dryad under the DOI: https://doi.org/10.5061/dryad.3j9kd51jb. Retrograde labeling data has been deposited in Dryad under the DOI: https://doi.org/10.5061/dryad.6q573n60c.

The following dataset was generated:

| Author(s) | Year | Dataset title | Dataset URL | Database and Identifier |
|---|---|---|---|---|
| Luft M, Klepetko J, Muceli S, Ibáñez J, Tereshenko V, Festin C, Längle G, Politikou O, Maierhofer U, Farina D, Aszmann O, Bergmeister K | 2021 | Muscle mass of the long head of the biceps following single and double nerve transfer | https://doi.org/10.5061/dryad.3j9kd51jb | Dryad Digital Repository, 10.5061/dryad.3j9kd51jb |
| Luft M | 2021 | Labeled motor neurons following single and double nerve reinnervation of the long head of the biceps | https://doi.org/10.5061/dryad.6q573n60c | Dryad Digital Repository, 10.5061/dryad.6q573n60c |

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
