## [Decision Letter]

**Acceptance summary:**

This paper will be of interest to neuroscientists and rehabilitation engineers who perform experiments involving peripheral innervation of muscles. It develops a new model for double nerve transfers in the forelimbs of rodents, and is therefore likely to be of great utility as a model for a range of studies related to nerve injury and therapeutics. It provides extensive guidance on how to reproduce this model, as well as detailed methods to characterize the success of neuromuscular regeneration.

**Decision letter after peer review:**

Thank you for submitting your article "Double nerve transfer to a single target muscle: experimental model in the upper extremity" for consideration by *eLife*. Your article has been reviewed by 3 peer reviewers, including Samantha R Santacruz as Reviewing Editor and Reviewer #1, and the evaluation has been overseen by Tamar Makin as the Senior Editor. The following individual involved in review of your submission has agreed to reveal their identity: David Lumenta (Reviewer #3).

Essential revisions:

1) Elaborate on the retrograde labeling results. We suggest to include results of differences of retrograde labeling, comparison of single vs. double nerve transfers (eg. labeling single nerve vs. only one branch of corresponding double nerve transfer).

2) Provide additional details on statistical analyses in the Methods. In particular, provide qualification to demonstrate that parametric tests (e.g. ANOVA, ANCOVA) were appropriate when used.

3) Present qualitative data in the Results section in the form of graphs and tables.

*Reviewer #1 (Recommendations for the authors):*

I have no additional suggestions for edits to the submission.

*Reviewer #2 (Recommendations for the authors):*

It would be great to better highlight the utility of multiples NTs in the introduction. Perhaps re-position some of the Discussion section items around clinical application into the introduction.

A supplemental table or brief figure demonstrating the most useful nerve donors/targets for 3-4 high value applications would be useful for the reader to get a sense of how and where this could be performed.

This technique could be applied to regenerative peripheral nerve interfaces (RPNIs) in which muscle mass is currently constrained by the axonal count, limiting EMG output for neuroprosthetic control. For instance, consider applications to the current work of Hugh Herr and Cindy Chestek.

Figure 1. looks beautiful, but most of the area is not being used as efficaciously as it could be. The zoomed in box is the most useful and could be made larger with greater details noted.

Title could better indicate the advantages of the DNT.

Qualitative data needs to be presented in the Results section in the form of graphs and tables.

*Reviewer #3 (Recommendations for the authors):*

Suggest to:

– Include results of differences of retrograde labeling, comparison of single vs. double nerve transfers (eg. labeling single nerve vs. only one branch of corresponding double nerve transfer).

– Explain ANCOVA statistical analysis (methods) to readers not familiar with the term.

– Add more details to statistical analysis (only in methods) to illustrate that you performed testing for parametric distribution.

---

## [Author Response]

Essential revisions:(1) Elaborate on the retrograde labeling results. We suggest to include results of differences of retrograde labeling, comparison of single vs. double nerve transfers (eg. labeling single nerve vs. only one branch of corresponding double nerve transfer).

Thank you for your suggestion. We included quantitative results and comparisons on the number of reinnervating motoneurons following SNT, DNT and data of a control group which assessed the number of axons innervating the untreated biceps’ long head (Please see section 2.3 of results p4 line 122-124, 129-137 and Figure 1C, section 3 of discussion p7 line 230-235, section 4.6. of Materials and methods p15 line 454-458). The Retrograde labeling data has been deposited in Dryad under the DOI: https://doi.org/10.5061/dryad.6q573n60c.

2) Provide additional details on statistical analyses in the Methods. In particular, provide qualification to demonstrate that parametric tests (e.g. ANOVA, ANCOVA) were appropriate when used.

Based on your suggestions, our biostatistician included more details about model diagnostics and whether the requirements for the parametric models were met. Please see section 4.6. Methods p15 line 459-465 and p4 line 129-137.

3) Present qualitative data in the Results section in the form of graphs and tables.

Thank you for your suggestion, which we included in a summarizing table presenting qualitative data see Section 2, Table 1. In addition, we present the results of the muscle masses and retrograde labeling in additional figures, as we think that this contributes to a better understanding and overview for the reader. Please see Figures 1C and 2.

Reviewer #2 (Recommendations for the authors):It would be great to better highlight the utility of multiples NTs in the introduction. Perhaps re-position some of the Discussion section items around clinical application into the introduction.

Thank you for the advice, we repositioned clinical application items from the discussion to the introduction and added some information about the clinical opportunities of multiple NT’s. Please see p. 3 line 85-89.

A supplemental table or brief figure demonstrating the most useful nerve donors/targets for 3-4 high value applications would be useful for the reader to get a sense of how and where this could be performed.

We agree and are currently working on a clinical project that will highlight the potential clinical use and how and where double nerve transfers can be functionally utilized. For this purpose, we are currently performing cadaver studies.

This technique could be applied to regenerative peripheral nerve interfaces (RPNIs) in which muscle mass is currently constrained by the axonal count, limiting EMG output for neuroprosthetic control. For instance, consider applications to the current work of Hugh Herr and Cindy Chestek.

Thank you, we included the work of Hugh Herr and Cynthia Chestek and referenced their publications, e.g. (Vu et al., 2020) and (Srinivasan et al., 2017). Also, we implemented considerations of combining the DNT with the RPNIs and agonist-antagonist myoneural interface (AMI) in the discussion. Please refer to p8 line 260-269.

Figure 1. looks beautiful, but most of the area is not being used as efficaciously as it could be. The zoomed in box is the most useful and could be made larger with greater details noted.Title could better indicate the advantages of the DNT.

Thank you for acknowledging the quality of the figure. We revised the figure according to your suggestion.

Qualitative data needs to be presented in the Results section in the form of graphs and tables.

Thank you for your suggestion, which we included in the essential revision 3.

Reviewer #3 (Recommendations for the authors):Suggest to:– Include results of differences of retrograde labeling, comparison of single vs. double nerve transfers (eg. labeling single nerve vs. only one branch of corresponding double nerve transfer).

Thank you for your suggestion. Please see the answer to essential revision 1.

– Explain ANCOVA statistical analysis (methods) to readers not familiar with the term.

Thank you for this request. We added a detailed explanation of what an ANCOVA is and why it was conducted here. Please refer to section 4.6. p15.

– Add more details to statistical analysis (only in methods) to illustrate that you performed testing for parametric distribution.

Please refer to the answer for essential revision 2.

References

Katada, A., Vos, J.D., Swelstad, B.B., and Zealear, D.L. (2006). A sequential double labeling technique for studying changes in motoneuronal projections to muscle following nerve injury and reinnervation. Journal of Neuroscience Methods 155, 20-27.

Srinivasan, S.S., Carty, M.J., Calvaresi, P.W., Clites, T.R., Maimon, B.E., Taylor, C.R., Zorzos, A.N., and Herr, H. (2017). On prosthetic control: A regenerative agonist-antagonist myoneural interface. Sci Robot 2.

Vu, P.P., Vaskov, A.K., Irwin, Z.T., Henning, P.T., Lueders, D.R., Laidlaw, A.T., Davis, A.J., Nu, C.S., Gates, D.H., Gillespie, R.B., Kemp, S.W.P., Kung, T.A., Chestek, C.A., and Cederna, P.S. (2020). A regenerative peripheral nerve interface allows real-time control of an artificial hand in upper limb amputees. Sci Transl Med 12.